# Neural Priming for Sample-Efficient Adaptation

**Matthew Wallingford**[*†]    **Vivek Ramanujan**[*†]
**Alex Fang**[†]    **Aditya Kusupati**[†]    **Roozbeh Mottaghi**[†]    **Aniruddha Kembhavi**[◇]
**Ludwig Schmidt**[†◇‡]    **Ali Farhadi**[†]
[†]University of Washington    [◇]PRIOR, Allen Institute for AI    [‡]LAION
{mcw244,ramanv}@cs.washington.edu

## Abstract

We propose Neural Priming, a technique for adapting large pretrained models to distribution shifts and downstream tasks given few or no labeled examples. Presented with class names or unlabeled test samples, Neural Priming enables the model to recall and conditions its parameters on relevant data seen throughout pretraining, thereby priming it for the test distribution. Neural Priming can be performed at inference, even for pretraining datasets as large as LAION-2B. Performing lightweight updates on the recalled data significantly improves accuracy across a variety of distribution shift and transfer learning benchmarks. Concretely, in the zero-shot setting, we see a $2.45\%$ improvement in accuracy on ImageNet and $3.81\%$ accuracy improvement on average across standard transfer learning benchmarks. Further, using Neural Priming at inference to adapt to distribution shift, we see a $1.41\%$ accuracy improvement on ImageNetV2. These results demonstrate the effectiveness of Neural Priming in addressing the challenge of limited labeled data and changing distributions. Code is available at https://github.com/RAIVNLab/neural-priming.

## 1   Introduction

Humans have a vast store of prior experience which we draw on to flexibly perform a diverse range of tasks [20, 5, 4, 12]. While engaging in an activity, we naturally retrieve relevant information or schema in a cognitive phenomena known as Priming [34]. This process ensures that necessary knowledge is readily accessible in memory, leading to enhanced performance for the task at hand [43]. Pre-trained, general-purpose models such as CLIP [40] and ALIGN [23] have extensive prior knowledge learned from large-scale, diverse datasets. These datasets seek to capture all natural variation in real data within their distribution. Can these models also benefit from something like priming? We observe that models trained even on the largest of such datasets often substantially improve in performance when fine-tuned on task-specific data. This begs the question of what the model learns from fine-tuning on the target dataset, if it already trained on many similar examples during pre-training.

We speculate that the effect of fine-tuning a pre-trained model on task-specific data is similar to that of priming. Given the sheer size and diversity of the pre-training dataset it becomes challenging for the model to find a consistent solution that is optimal for all subsets of the data. This becomes particularly evident for open-vocabulary models such as CLIP, where multiple natural language descriptions can correspond to a single image, highlighting the challenge of accommodating diverse interpretations. We hypothesize that training on the downstream dataset re-aligns the model to the specific objective.

With this in consideration, we propose Neural Priming. Specifically, Neural Priming recalls a subset of the pre-training data similar to the target distribution, re-aligns the natural language

---

[*]Equal contribution

37th Conference on Neural Information Processing Systems (NeurIPS 2023).

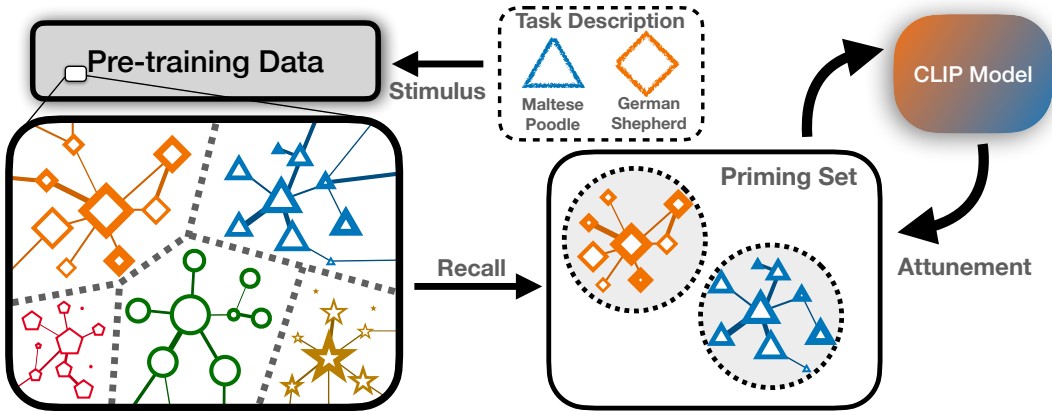

Figure 1: **A diagram of** Neural Priming**, our proposed method.** Neural Priming is a framework for leveraging an open-vocabulary model's *own pre-training data* to improve performance on downstream tasks. Neural Priming encompasses two processes: **1.** Collecting a *priming pool* of relevant examples from the pre-training set to prime with and **2.** using these examples to attune our model to a given task. We show performance improvements across a wide range of transfer learning and robustness benchmarks.

descriptions to the downstream task, and quickly adapts the model to the subset. We perform extensive experiments on 7 transfer learning and 4 distribution shift datasets to validate our method. We use the OpenCLIP [40, 52] ViT [8] set of models pre-trained on the LAION-2B and 400M [44]. We find Neural Priming leads to significant accuracy improvements, particularly when labeled data is scarce and in specialized domains. Concretely, Neural Priming improves accuracy by 2.45% on ImageNet and 4.25% on average across the other 6 datasets over the base CLIP model. In the few-shot setting, Neural Priming improves accuracy by 3.81% on average over recent methods on standard transfer learning benchmarks. We show Neural Priming is efficient and can be performed on-the-fly. For datasets containing more than 2 billion images, we can prime our model to ImageNet in less than 2 minutes with a single commercial GPU.

Neural Priming is flexible and can be used with variable degrees of information about the downstream distribution. When the model has language-only task descriptions, our approach can efficiently retrieve a *priming pool* of relevant examples from the pre-training set and attune the model to this data. At inference time, given a set of test images to classify, Neural Priming is able to use test examples to adapt the priming pool to distribution shifts. When we have access to training examples in the few-shot setting, Neural Priming can filter the priming pool to align with the training distribution.

**We make the following contributions:**

- We introduce Neural Priming, a novel method that leverages retrieval from the pre-training dataset for efficient and accurate adaptation of large, pre-trained models to downstream tasks.
- Neural Priming achieves state-of-the-art zero-shot and few-shot accuracy on the standard transfer learning datasets – up to 4.25% and 3.81% improvements respectively over baselines (Section 4.2).
- Neural Priming also enables transductive learning and improves performance on standard distribution shift datasets by 2.51% on average, all without using any additional data (Section 4.3).
- Our approach generalizes to various architectures and pre-training datasets while being complementary to techniques [33, 39] that improve zero-shot performance of open-vocabulary models.

## 2 Related Work

### 2.1 Open-Vocabulary Models and Zero-shot Inference

Open-vocabulary models have proven to be an effective approach for transfer learning. Such models enable training on vast amounts of web-scale images without the need for labor-intensive human

labeling by leveraging pre-existing natural language descriptions [40]. Open-vocabulary models have set state-of-the-art on ImageNet [7] as well other transfer learning benchmarks [53, 1, 41].

Open-vocabulary models offer additional capabilities beyond standard pre-trained models. They can perform zero-shot inference, where predictions are made without training on target data. Additionally, they are robust to distribution shifts [40], enable prompt-tuning methods [39, 33], and can be used for text-based retrieval [10]. Zero-shot can have different meanings in the literature. In the context of this paper, we consider zero-shot as the experimental setting in which the model receives no training examples drawn from the training distribution.

Prompt-tuning has emerged as a popular research direction in the domains of large language and open-vocabulary models. In the context of open-vocabulary models, prompt-tuning can involve modifying the textual prompts or queries used during the training or inference of the model to improve its understanding of visual content or achieve specific goals. In the original CLIP paper, Radford et al. [40] design hand-crafted prompt templates for ImageNet and other transfer learning datasets and show that this leads to substantial accuracy improvements. More recently, other work [33, 55] has used machine learning approaches to learn the prompts rather than hand-crafting them.

## 2.2 Distribution Shifts

Robustness to distribution shift is a key property of good machine learning models as it represents a notion of reliability. In particular, studies on natural distribution shifts, including ImageNet-V2 [41], ImageNet-Sketch [51], ImageNet-R [18], and ImageNet-A [19], find that models have a consistent performance drop when exposed to a distribution at inference time not seen during train time [49]. In order to focus on robustness and eliminate the confounder of better models being generally better, this performance gap is measured through effective robustness, which is the robustness improvement over ImageNet trained models. Prior work has shown that the performance of models on in distribution and out of distribution is highly correlated across many algorithmic training interventions, except for cases where training on larger and more diverse datasets increases robustness [35].

The most significant recent improvement in effective robustness [41] is the introduction of open-vocabulary models. At its time of release, CLIP [40] achieved unprecedented effective robustness on a variety of distribution shifts. Studies have suggested that these models achieve high effective robustness through their data distribution [9], a result of training on large amounts of web-scraped data. However, these models are still worse at downstream tasks than models fine-tuned on in-distribution data. Moreover, fine-tuning on downstream data causes robustness on other data distributions to deteriorate [40, 52]. Many mitigation methods have been proposed to such as Wise-FT, FLYP, LP-FT, and model surgery [52, 15, 28, 29]. Our paper differs from these methods in goal: whereas they seek to keep model robustness while gaining the benefits of fine-tuning on task-specific data, we seek the benefits of fine-tuning while *not collecting any in-distribution data*. Hence these methods are complementary to Neural Priming, and we employ Wise-FT in our model attunement procedure.

## 2.3 Transductive Learning

Transductive learning [13, 6] focuses on leveraging unlabeled data during inference. It differs from traditional supervised learning, which solely relies on labeled data at train time. Related to transductive learning is test-time training [48, 14, 45]. Test-time training involves adapting and refining the model's predictions based on the specific testing examples encountered. Transductive learning differs from test-time training in that test-time training only considers one test sample at a time, whereas transductive aims to learn from the entire test set.

## 2.4 Few-Shot Learning

Few-shot learning research aims to addresses the challenge of learning from a limited number of labeled examples. In many real-world scenarios, acquiring large labeled datasets is impractical or costly. Older lines of work have focused on meta-training small models [47, 11, 37, 22] on small-scale datasets. More recently, the approach for few-shot learning has shifted towards training large, general-purpose models such as CLIP [40] and ALIGN [23] on web-scale datasets.

## 2.5 Retrieval-Augmented Models

In language, works have demonstrated the effectiveness of retrieval from text corpora or structured data for tasks such as question answering [3, 16, 25]. In general, these methods seek to recover facts either from a large corpus or knowledge graph, then use those to complete tasks. This differs from our scenario, where exact examples at inference time do not necessarily exist in the pre-training corpus. REACT [30] and SuS-X [50] are retrieval-augmented methods for open-vocabulary models which use search to fine-tune with relevant examples [30]. We differ from Liu et al. [30] in that they add a substantial number of new parameters whereas we do not. Additionally, our approach is significantly more efficient, both computationally and in terms of number of samples, enabling use at inference for additional improvement (Section 3.1.2). We differ from [50] in that their work uses semantic retrieval whereas Neural Priming leverages language for fast initial filtering and image search for accurate retrieval. Further, Neural Priming shows that models can improve by revisiting examples seen throughout pretraining whereas other works retrieve new examples from external datasets.

## 3 Method

Neural Priming is the process of retrieving relevant information from the pre-training dataset and leveraging it for a specific task. We study it in the context of vision-language contrastive pre-training, so the form our task description takes is a set of class names, $\mathcal{C}$, already in natural language. A CLIP model [40] consists of a vision embedding model, $V$, and a language embedding model, $L$, each producing a vector representation in $\mathbb{R}^d$. The pre-training dataset, $\mathcal{D}$, consists of a large number of image-text pairs collected from the web. The text component can be noisy, potentially containing irrelevant or inaccurate information about the image content.

We break our method down into two main steps: **1.** Collecting the priming pool, where we gather data from our pre-training dataset relevant to a particular task and **2.** model attunement, where we leverage this data to improve our model.

### 3.1 Collecting the Priming Pool

#### 3.1.1 Leveraging Natural Language Task Information

The goal of this step is to collect an initial pool of images relevant to the task at hand given the previously defined natural language description $\mathcal{C}$. For example, if our task is a set of dog breeds, ideally we would collect sets of images belonging to those breeds and label them accordingly. A simple way to prime is by using retrieval to gather relevant data points from our pre-training dataset. An existing method for language-based retrieval involves using the CLIP text embedding of a class description $c \in \mathcal{C}$ for retrieval using semantic similarity scores on the pre-training set [2, 42]. However, with neural priming, prioritizing precision over recall is crucial, considering the size, diversity, and noise of the pre-training dataset. This form of semantic retrieval has a major downside: it is not clear where to threshold similarity scores to retrieve the most relevant images. Threshold too late and we allow unrelated images to be included in our pool. Further, this threshold is often specific to a category, making it infeasible to search at scale.

Our approach to language-based priming is to search for the existence of the class name, $c \in \mathcal{C}$, in the captions of our pre-training dataset to retrieve images relevant to a particular category. We organize these image-text pairs into separate categorical clusters $\{B_c\}$ according to the class name $c$ mentioned in their captions. This approach has a few advantages over semantic retrieval: **1.** After setting up an inverted index search structure for text retrieval [26, 46], exact string matching is far faster than semantic retrieval, even when approximate nearest neighbor strategies are employed [24], **2.** with exact string search, the category boundary is clear and therefore does not require per category tuning, **3.** the retrieval results are overall qualitatively more relevant. Finally, to leverage the semantic understanding of our CLIP model, we filter the priming pool using CLIP similarity score. We do this by constructing a "zero-shot" CLIP classifier, as defined by Radford et al. [40], and removing examples from categorical clusters that do not align with their label according to the CLIP model.

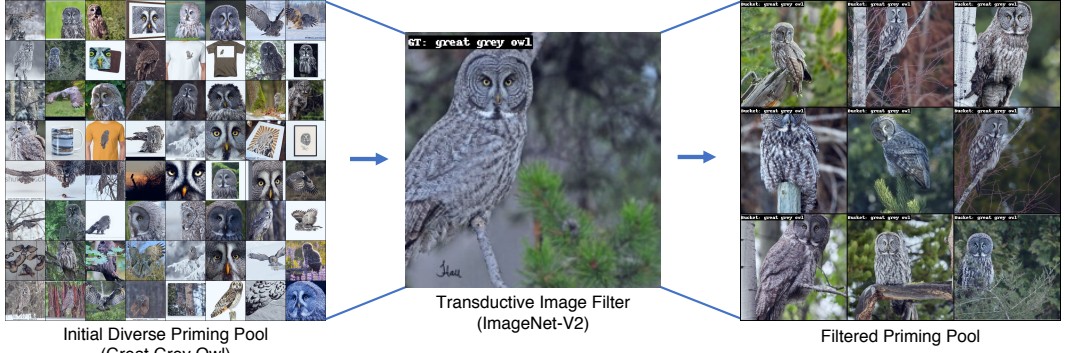

Initial Diverse Priming Pool
(Great Grey Owl)

Transductive Image Filter
(ImageNet-V2)

Filtered Priming Pool

Figure 2: **A qualitative example of our approach for transductive image filtering.** Given an initial *priming pool*, acquired through natural-language text search on the captions of our pre-training dataset (Section 3.1), we filter out irrelevant examples using images from our test set. **(left)** we show examples from the great owl categorical cluster of our priming pool before filtering, **(center)** we show an example image from the same category of ImageNet-V2, **(right)** example retrievals using image embedding similarity from the entire priming pool. The visual similarity of the retrievals are apparent, and they are generally from the appropriate categorical cluster. Doing this filtering results in a significantly more relevant priming pool.

### 3.1.2 Leveraging Image Information at Test Time

At inference time, the model can narrow the relevant priming pool even further by utilizing information about the test distribution. To do this, given an image $x$ in our test set, we compute the cosine similarity using our CLIP image encoder $V$, $\cos(V(x), V(y))$ for every $y \in P$, our priming pool. We retrieve examples with the top-$k$ cosine similarity scores ($k = 10$ in most of our experiments). We do this collectively for every image in the test set and collect the retrievals to form a filtered priming pool. If an example is retrieved twice, we de-duplicate them in the final priming pool. Since we do this for all images in the test set, we consider this the *transductive setting* to align with prior work [13, 6].

### 3.2 Attuning CLIP to the Priming Pool

The goal of this step is to modify our CLIP model to take advantage of the data in the priming pool $P$. We first construct the task-specific zero-shot linear head $W_z \in \mathbb{R}^{d \times n}$ using the text encoder and the natural language names of each class, where $d$ is the feature dimension and $n$ is the number of classes. To get logits for a particular example, $x$, we compute $W_z \cdot L(x)$, so our prediction is $\arg\max_c W_z \cdot L(x)$.

To attune our CLIP model to the priming pool, we perform nearest-class mean (NCM) [32] on all retrieved examples to obtain a classification head from the priming pool. Namely for a given class $c$, we compute a centroid $\tilde{y}_c = \frac{1}{|B_c|} \sum_{x \in B_c} L(x)$ and then normalize this centroid to produce a class embedding $y_c = \tilde{y}_c / \|y_c\|$. We define the collection of centroids as matrix $W_{ft} = [y_c]_c \in \mathbb{R}^{d \times n}$. To expand this to few-shot scenarios, we mix the labeled data into the corresponding categorical clusters before performing NCM. Finally, we ensemble $W_z$ and $W_{ft}$ using a mixing coefficient $\alpha \in [0, 1]$ as $W_\alpha = (1 - \alpha) \cdot W_{ft} + \alpha \cdot W_z$, which is our final classification head. We choose alpha according to a heuristic $\alpha = e^{-|P|/\sigma^2}$ which can be derived from a Bayesian prior over the text features. We experiment with varying values of $\sigma$ in Table 10. We also find that $\alpha$ can be effectively chosen through cross-validation on a held-out portion of the retrieval set. Intuitively, if we do not have much data in our priming pool, we want it to influence our model less. We use NCM as the classifier as it has shown to be sample-efficient [47]. For comparison to a WISE-FT classifier [52] see Table 3.

## 4   Experiments

Our key results include: **1.** Priming improves performance over baselines in the few-shot setting by 3.81% on average across all datasets and 2.4% on ImageNet in the zero-shot setting **2.** Priming in the

Table 1: **Performance of Neural Priming and comparable methods in the zero-shot setting.** Priming consistently improves top-1 accuracy across standard transfer learning data sets. Performance reported for the OpenCLIP ViT-B-16 model pretrained on LAION-2B.

|  | ImageNet | Stanford Cars | FGVC Aircraft | Flowers102 | Food101 | Oxford Pets | SUN397 |
|---|---|---|---|---|---|---|---|
| CLIP [40, 21] | 68.30 | 87.40 | 25.86 | 71.65 | 86.58 | 90.21 | 67.35 |
| Retrieval + Finetuning | 70.28 | 87.95 | 26.22 | 72.15 | 86.63 | 90.35 | 68.01 |
| VLM [33] | 69.35 | 87.88 | 28.54 | 72.11 | 86.31 | 90.24 | 67.73 |
| CuPL [39] | 70.25 | 88.63 | 29.64 | 72.32 | 86.20 | 91.16 | 70.80 |
| Priming (Ours) | 70.75 | 89.30 | 33.03 | 79.81 | 86.66 | **91.87** | 71.21 |
| Priming + CuPL (Ours) | **71.38** | **90.23** | **36.00** | **80.04** | **86.86** | 91.85 | **72.35** |

transductive, or on-the-fly, setting further improves performance over baselines by 2.51% accuracy and 1.09% over standard Neural Priming **3.** Priming is complementary to existing prompt-tuning methods. Our finding indicates that images in the priming set impart distinct information to the model compared to textual class descriptions. We include full details of hyperparameter choices and error bars included in the appendix.

## 4.1 Datasets and Architectures

We evaluate on standard transfer learning and distribution shift benchmarks. ImageNet [7] is a large-scale, general classification dataset that has been well-studied in both transfer learning and distribution shift. ImageNetV2 [41] is one of its natural distribution shift test sets, made by reproducing the original data collection procedure of ImageNet, but even modern large-scale pre-trained models have performance drops on it. ImageNet Sketch [51] and ImageNet-R [18] are natural distribution shifts created by assembling sketches and various renditions of the ImageNet classes. ImageNet-A [19] is a natural adversarial distribution shift of ImageNet, created by collecting images that are misclassified by ResNets. StanfordCars [27], FGVCAircraft [31], Flowers102 [36], and OxfordPets [38] are fine-grained classification datasets which require understanding subtle visual differences between classes and are commonly used for transfer learning benchmarks [17, 21, 40, 55]. SUN397 [53] is a large-scale scene recognition dataset with 397 scene categories.

We perform our experiments with OpenCLIP models [52] trained on LAION-2B and 400M [44]. We choose OpenCLIP because their pretrain datasets are publicly available, therefore we can control what data is introduced to the model. The model architecture reported in the main paper is the B-16 variant trained on LAION-2B unless otherwise stated and we report L-14 and B-32 in the Appendix C.

## 4.2 Zero-shot Results

In this setting, our model only has access to data it has seen during pre-training, in this case LAION-2B. Neural Priming improves top-1 accuracy by 2.45% on ImageNet and 4.25% on average across 6 other diverse downstream datasets compared to the CLIP baseline. In the zero-shot setting, Neural Priming outperforms the 3-shot CLIP model on StanfordCars and FGVCAircraft. This result is particularly noteworthy since traditionally training on in-distribution data generally outperforms zero-shot techniques [54]. Note that we do not present error-bars for the zero-shot experiments as the process is deterministic.

We also compare with VLM [33] and CuPL [39], two zero-shot prompt-tuning methods which obtain natural language descriptions of each class using language models, and a retrieval with fine-tuning baseline. For implementation details on how the retrieval and fine-tuning are performed see Appendix H. Interestingly, we find that Neural Priming is complementary to existing prompt-tuning methods. The accuracy improvements from CuPL and VLM are additive with Neural Priming. For example, CuPL and Neural Priming each improve performance by **3.78%** and **7.17%** respectively on FGVCAircraft. Ensembling the methods results in **10.74%** improvement over the baseline (Table 2). This surprising result suggests that the textual class descriptions in CuPL and VLM provide unique information to the model that differ from the information obtained from the images in the priming set.

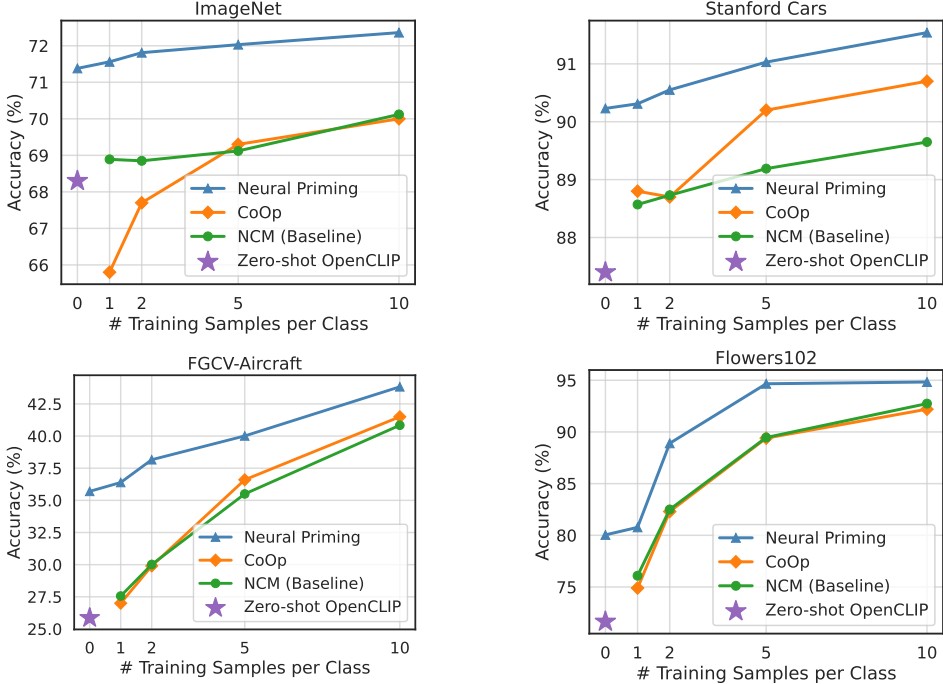

Figure 3: **Performance of Neural Priming and comparable methods in the few-shot setting.**
We find consistent improvement across shot numbers and datasets. In particular, Neural Priming
especially excels for fine-grained datasets such as FGVCAircraft and Flowers102. We hypothesize
that such fine-grained captioned images are not well represented in LAION-2B, therefore revisting
this subset of data improves the model more.

Another observation is that Neural Priming is especially effective for specialized domains such as
StanfordCars, Flowers102, and FGVCAircraft. We speculate this is due to the fact that the label
space and image content differs from the majority of the pre-training set. For example, although
airplanes occur frequently in LAION-2B, they are rarely described according to their specific model
such as *Boeing 737-200*. Therefore, recalling and priming the model on pre-train images with such
fine-grained classes significantly improves the model. For analysis of LAION-2B with regards to
label statistics see Appendix B.

In contrast, for datasets which are more aligned with LAION-2B and the distribution of internet
images, such as ImageNet and SUN397, the accuracy gain provided from Neural Priming is smaller
in comparison, albeit still significant. In the limit of this trend, Food101 sees almost no improvement
across all methods, and even training on in-distribution data for the few-shot case barely improves
the accuracy. We speculate that this is because images similar to those in Food101 are already well-
represented in LAION-2B, rendering additional food images of marginal informational value. We
provide analysis of how well the attributes of each dataset are captured by LAION-2B in Appendix B.

To be precise, when we refer to term "shot number" throughout the experiments section, we mean the
number of labeled examples from the target training set. We do not consider images retrieved from
LAION-2B as shots in this setting because they are obtained from the pre-training set.

### 4.3 Few-Shot Results

Neural Priming improves performance for all datasets and shots in the few-shot setting. We compare
with CoOp, a recent method for few-shot prompt-tuning, and a Nearest-class-Mean (NCM) baseline.
On average across all shots and datasets Neural Priming improves by 3.81% in accuracy over the
closest baseline. Results can be found in Figure 3 and Table 9 of the Appendix.

Notably, we find that Neural Priming can match the accuracy of models trained with a substantial
number of training examples *without using any of the labeled training data* for all of the evaluated

Table 2: **Performance of Neural Priming and relevant methods for the transductive setting.** Neural Priming finds examples similar to the test image at inference to optimize the model. Models are evaluated zero-shot on 4 distribution shift datasets. Neural Priming excels on distribution shifts which differ significantly from the natural language description of the class names. Performance reported for the OpenCLIP ViT-B-16 model pretrained on LAION-2B.

|  | ImageNet-V2 | ImageNet-R | ImageNet Sketch | ImageNet-A |
|---|---|---|---|---|
| CLIP [21, 40] | 59.35 | 64.57 | 57.05 | 35.95 |
| TPT [45] | 59.84 | 78.74 | 52.75 | 36.92 |
| Priming (Ours) | 60.12 | 77.98 | 58.29 | 37.56 |
| Transduct. Priming (Ours) | **60.76** | **79.37** | **59.97** | **38.20** |

datasets (Figure 3). Similar to the zero-shot setting, we observe that Neural Priming is complementary with prompt-tuning methods (Appendix F). Additionally, we observe that as the shot number increases, improvement over the baseline decreases. At 1-shot the improvement in accuracy over the baselines is 5.63% on average, while at 10-shot the improvement is 2.04%. Intuitively, as the model receives more target training data, obtaining additional examples from the pretrain set becomes less necessary.

## 4.4 Transductive Results

We compare Neural Priming in the transductive setting on 4 standard distribution shift datasets, ImageNet-V2, ImageNet Sketch, ImageNet-R and ImageNet-A. Distribution shift datasets are a natural application of adaptation at test-time. Often real-world datasets differ from the training data, therefore models should be able to adapt on-the-fly. In this setting, the model can learn from the test images without labels before making predictions. We compare with Test-Time Prompt-Tuning (TPT), a state-of-the-art method which uses a self-supervised objective to learn from test data.

We find that Neural Priming with images in the test set improves performance over standard Neural Priming by 1.09% as well as 2.51% over TPT across the 4 distribution shifts (Table 2). Looking at Figure 2, we qualitatively see that the priming pool more closely matches the test images after filtering for the closest images in the initial priming pool. Though the distribution shift can often be imperceptible such as between ImageNet and ImageNetv2, quantitatively we see that the transductive filtering step finds images in the pretraining close to the test distribution.

The transductive retrieval for 50,000 images in the test set on average takes 96 seconds for a priming pool of 1 million images, while retraining the classifier takes on average 11.5 seconds for a priming pool of size 10,000 on standard hardware. We provide further analysis of run-time efficiency of the on-the-fly variant of Neural Priming in Appendix G.

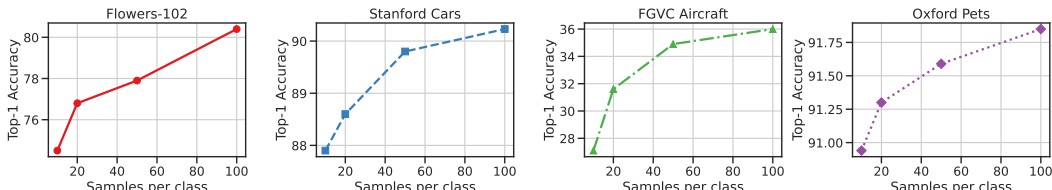

Figure 4: **Ablation over the number of samples per class in the priming pool.** We observe a consistent zero-shot accuracy improvement as the number of samples drawn from our pool increases.

## 4.5 Ablations

We investigate the impact of the priming pool size on the zero-shot accuracy of downstream tasks (Figure 4). Our analysis reveals that as the size of the priming pool increases, there is a general improvement in accuracy. However, there are certain limitations associated with enlarging the pool. The majority of classes in the downstream task have a limited number of available images.

Consequently, when we retrieve a larger number of images for the priming pool, they tend to contain more noise and mislabeled samples. Furthermore, for rare classes, the number of images obtained through exact string search is often less than 100. To address this, a potential extension could involve utilizing a language model to generate alias names for classes, which could then be used to perform additional string searches, thereby expanding the initial priming pool size.

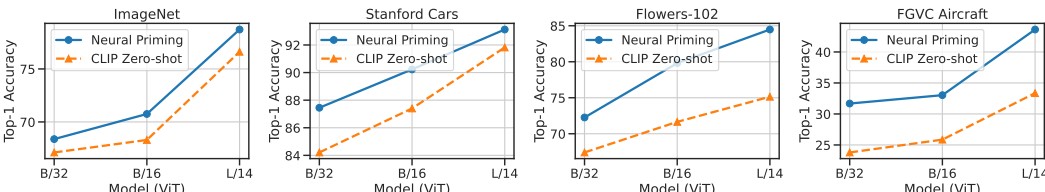

Figure 5: **Analyzing the effect of model capacity on Neural Priming.** We find the relative error reduction stays consistent even as the scale of the model increases.

We also analyze the impact of the architecture on the accuracy improvement achieved by Neural Priming in the zero-shot setting (Figure 5). To examine this, we conduct experiments using models of varying capacities, namely ViT B-32, B-16, and L-14. We observe that the gains remains consistent across the models. This finding suggests that even as we scale the architecture's capacity, our method will continue to yield significant and consistent relative error reduction.

## 5 Limitations

Neural Priming has a few potential limitations. Firstly, it requires that the pre-train dataset contains images similar to those in the downstream task. Though all of the datasets we benchmark have abundant relevant data, it is possible for more out-of-distribution datasets that LAION-2B simply does not contain related or queryable images. Secondly, accurate class names are required for retrieval. Meaningful class names for some datasets can be difficult to obtain. For example, in the Flowers102 dataset, some flower species are given by their latin names, which leads to poor retrieval. This issue generally affects open-vocabulary models which require accurate class names to initialize the zero-shot classifier. This limitation may be resolved by using language models to replace class names with their more commonly known synonyms. Lastly, Neural Priming requires access to the pre-training data set which is not always possible such as in the case of OpenAI variant of CLIP. In this case a surrogate dataset would likely suffice, such as using LAION-2B.

## 6 Discussion & Conclusion

We present Neural Priming, a method to improve the performance of open-vocabulary models by leveraging their own large-scale, diverse pre-training data with no additional data required. With Neural Priming, we demonstrate how to construct a high quality priming pool of examples from the pre-training dataset relevant to a particular task and how to utilize this pool to improve our model. We further show that our method is effective across a variety of downstream tasks and settings. In particular, our method can be used in situations where only natural language descriptions of relevant classes are given, when we have the ability to adapt at inference time, and when we are provided with few labeled in-distribution examples. In all settings, our framework demonstrates a substantial improvement in performance over existing interventions, and is in fact complementary with current prompt-tuning and robustness methods. Our method is also computationally cheap, not requiring any modification of model backbone weights and only a fast text search on the pre-training corpus.

The efficacy of Neural Priming leads to some interesting questions for future work. For example, if the model has seen this data before, why does it help to recall them? We hypothesize that this is due to the fact that the diversity of these datasets introduces competing objectives, which are difficult for the model to optimize directly. For example, the same kind of image could appear with multiple captions and vice-versa, making it difficult to prompt a CLIP model trained on such data at inference time for a particular task. A systematic study of this could elucidate important limitations of current large-scale training paradigms.

## Acknowledgments

We are grateful to Sarah Pratt, Mitchell Wortsman, and Romain Beaumont for helpful discussions and feedback. Ali Farhadi acknowledges funding from the NSF awards IIS 1652052, IIS 17303166, DARPA N66001-19-2-4031, DARPA W911NF-15-1-0543, and gifts from Allen Institute for Artificial Intelligence, Google, and Apple. Ludwig Schmidt and Alex Fang are in part supported by the NSF AI Institute for Foundations of Machine Learning (IFML, CCF-2019844), Open Philanthropy, Google, and the Allen Institute for AI.

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

# A   Implementation Details

We perform last layer retraining for Neural Priming. We experimented with fine-tuning all layers, but found minimal performance benefits (Table 4). It has been noted in few-shot literature [47, 37] that retraining only the last layer is sufficient when few training examples are present and we confirm this empirically. When retraining the last layer, we ensemble the text-classifier. We use a mixing co-efficient to ensemble the text-classifier with the image centroids. We found that as the number of samples in the image centroid increases, the ensemble should more heavily favor the image centroid. Formally we use $\alpha = e^{-|P|^2/\sigma}$ as the mixing-coefficient, where $|P|$ is the number of image examples with $\sigma$ equal to 100.

The OpenCLIP models we use can be found at https://github.com/mlfoundations/open_clip. We use the following specific models: (ViT-B-32, laion2b_s34b_b79k), (ViT-B-16, laion2b_s34b_b88k), (ViT-L-14, laion2b_s32b_b82k).

For the zero-shot CLIP baseline presented in Table 2, we use the baseline prompts from [33]. When ensembling with the image centroids from NCM, we use the text-classifier initialized with these prompts unless otherwise stated.

To perform fast substring search, we set up a Full Text Search (FTS) database over the metadata shards of LAION-2B using SQLite. Each shard is a SQLite database composed of 18.1 million rows consisting of information about the URL of the image location, constructed from the original distributed shards from LAION-5B [44]. The nature of FTS and storage reading attributes means that it is much faster to search for rare substrings. More details on this are provided in Appendix G.

Table 3: **Comparing Neural Priming to other classification approaches.** NCM uses only the image features. WISE-FT fine-tunes a linear classifier and ensembles it with the text classifier.

| Shots | Neural Priming | NCM | Wise-FT |
|-------|----------------|-------|---------|
| 0 | 71.38 | 68.3 | 67.75 |
| 1 | 71.56 | 68.89 | 68.14 |
| 2 | 71.81 | 68.85 | 68.25 |
| 5 | 72.03 | 69.12 | 68.78 |
| 10 | 72.36 | 70.12 | 69.02 |

# B   LAION-2B Analysis

We analyze the prevalence of each domain in the LAION pretrain set (Figure 6). We find that for datasets where the domain images and captions occur infrequently such as Flowers102 and StanfordCars, the performance gain from Neural Priming is significantly larger. We measure domain prevalence in LAION-2B by substring searching for captions which contain the class names and counting the total number of images. Image content is another factor for measuring how close the pretraining set is to the downstream dataset in feature space, however this is computationally expensive for a dataset of 2 billion. Also this would depend on the model used to extract features. We choose to look at the caption statistics as a proxy because it is computationally feasible with a dataset at the size of LAION-2B and model agnostic.

# C   Other Architectures and Datasets

We present the full results for ViT B/16, B/32 and ViT L/14 on LAION-2B for ImageNet, Flowers102, StanfordCars, FGVC-Aircraft, and OxfordPets. We find similar results across architectures. Results can be found in Table 6. We also reproduce experiments for the LAION-400m dataset with the B/16 architecture.

# D   Dataset Statistics

For ImageNet we performed URL deduplication using the LAION-2B URL list provided on HuggingFace. We found no ImageNet images in the pretrain data. We also randomly sampled images

Table 4: **Comparison of full fine-tuning and NCM with Neural Priming.** We find NCM performs similarly in the low-sample regime.

|  | ImageNet | Stanford Cars | FGVC Aircraft | Flowers102 | Food101 | Oxford Pets | SUN397 |
|---|---|---|---|---|---|---|---|
| CLIP [40, 21] | 68.30 | 87.40 | 25.86 | 71.65 | 86.58 | 90.21 | 67.35 |
| Retrieval + FT | 70.28 | 87.95 | 26.22 | 72.15 | 86.63 | 90.35 | 68.01 |
| VLM [33] | 69.35 | 87.88 | 28.54 | 72.11 | 86.31 | 90.24 | 67.73 |
| CuPL [39] | 70.25 | 88.63 | 29.64 | 72.32 | 86.20 | 91.16 | 70.80 |
| Priming (Ours) | 70.75 | 89.30 | 33.03 | 79.81 | 86.66 | **91.87** | 71.21 |
| Priming + CuPL (Ours) | **71.38** | **90.23** | **36.00** | **80.04** | **86.86** | 91.85 | **72.35** |

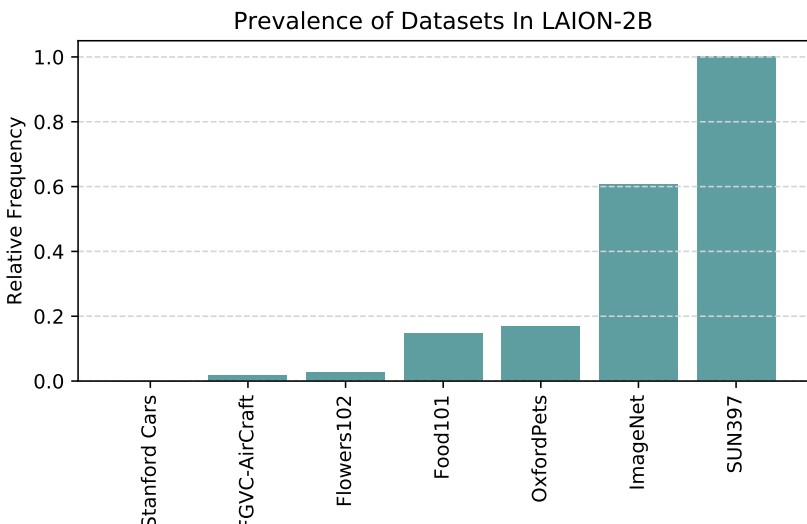

Figure 6: **Frequency of dataset class names in the LAION-2B dataset normalized to 1.** We find Neural Priming improves accuracy more for datasets that are rare.

from the other datasets (Flowers102, FGVCAirCraft, etc.) and looked for duplicates, though we found no exact matches. The exact dataset statistics are reported for context in Table 7.

# E  Comparison to REACT

We do not compare directly with REACT [30] in the main paper for two reasons. The first is that REACT adds a significant number of parameters (50 % for the B/16 model) to the network, making an unequal comparison from a parameter and FLOPS perspective. Second, REACT uses 6-10 million images as the support set for each dataset which makes it computationally infeasible to rerun ourselves for all of the data sets we benchmark on and without the parameter adding version. We compare on ImageNet using the numbers reported in their work for comparable models trained on LAION-2B (Table 8).

We differ from REACT in a few key aspects. First, we realign the images labels to the downstream task. For example, for the OxfordPets data set, given an image which has the caption "maltese on a leash walking through the parking with its owner" we map the label to "maltese". REACT trains contrastively using the original captions. We find this step significantly affects accuracy and sample-efficiency. For comparison, REACT retrieves 6-10 million images from LAION whereas we use 10,000-100,000. Second, our method is significantly more efficient in the retrieval and fine-tuning step so can be performed at test-time. This is due to our multi-stage filtering with the first stage of caption filtering being extremely fast. Third, REACT adds 50% additional parameters to the ViT B/16 model.

Table 5: **Performance of Neural Priming and the CLIP baseline with the ViT B/16 architecture.**
We observe similar trends as for the LAION-2B dataset. Neural Priming improves over the zero-shot
performance with similar relative error reduction.

|  | ImageNet | Stanford Cars | FGVC Aircraft | Flowers102 | Food101 | Oxford Pets | SUN397 |
|---|---|---|---|---|---|---|---|
| CLIP [40, 21] | 66.72 | 82.47 | 16.05 | 71.47 | 86.33 | 88.68 | 61.13 |
| Neural Priming (Ours) | 69.21 | 86.83 | 26.73 | 82.17 | 86.62 | 90.08 | 62.27 |

Table 6: **Zero-shot results for ViT B/32 and L/14 architectures.** We find similar results across
architectures. Surprisingly, the relative error often decreases more with architecture scale.

| Capacity | ImageNet | Stanford Cars | Flowers102 | FGVCAircraft | Food101 | Oxford Pets | SUN397 |
|---|---|---|---|---|---|---|---|
| | | | Baseline (NCM) | | | | |
| ViT-B/32 | 67.12 | 84.21 | 67.40 | 23.79 | 82.53 | 90.62 | 61.97 |
| ViT-B/16 | 68.30 | 87.40 | 71.65 | 25.86 | 86.58 | 90.21 | 67.35 |
| ViT-L/14 | 76.62 | 91.82 | 75.14 | 33.36 | 90.96 | 93.40 | 71.29 |
| | | | Neural Priming | | | | |
| ViT-B/32 | 68.38 | 87.45 | 72.26 | 31.68 | 82.91 | 91.25 | 63.84 |
| ViT-B/16 | 70.75 | 90.23 | 79.81 | 33.03 | 86.66 | 91.87 | 71.21 |
| ViT-L/14 | 78.72 | 93.12 | 84.48 | 43.62 | 91.06 | 94.05 | 72.49 |

# F    Few-Shot

We report the few-shot performance of a model pretrained on LAION-2B with CuPL prompts (Table
9). We find that better prompts synergize with Neural Priming similar to the zero-shot case. The
mixing coefficients used for these experiments are set by the schedule given in section A.

# G    Transductive Run-Time Analysis

Transductive run-time can be broken up into two components: **1.** Time to get the initial priming pool
and **2.** Time to perform exact retrieval on image features of the priming pool.

## G.1    Initial Priming Pool construction timings

Here we study the wall-clock time for construction of the initial priming pool on consumer hardware
(Intel 10900k CPU and Samsung 980 Pro NVME drive). Wall-clock times for priming pool construc-
tion vary greatly depending on a variety of factors. The two main factors affecting timings are: **1.**
speed of the storage on which the fast text search (FTS) databases are stored (see Appendix A) and
**2.** whether or not LAION-2b is stored locally (if it is not stored locally, then you need to access the

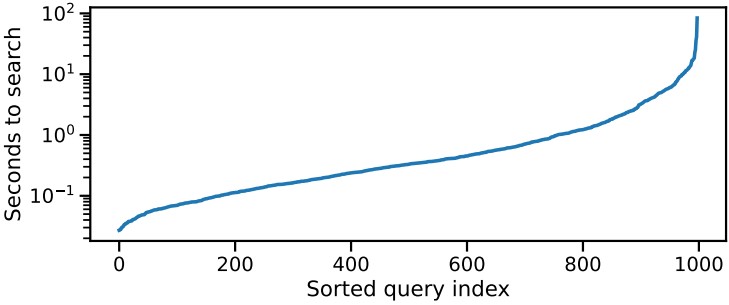

Figure 7: **Wall-clock times by query for construction of the initial priming pool.** The queries used
here are the ones used for the construction of the ImageNet pool. Due to the nature of full text search,
rare queries are very fast, with most classes taking less than 1 second to search all of LAION-2b.

| Dataset | Number of Classes | Train Size | Test Size |
|---|---|---|---|
| ImageNet | 1000 | 1281167 | 50000 |
| Stanford Cars | 196 | 8144 | 8041 |
| FGVC Aircraft | 100 | 6667 | 3333 |
| Flowers102 | 102 | 2040 | 6149 |
| Food101 | 102 | 75750 | 25250 |
| Oxford Pets | 37 | 3680 | 3669 |
| SUN397 | 397 | 19850 | 19850 |

Table 7: Dataset statistics for every dataset evaluated on in Section 4. We evaluated using the accuracy metric for every dataset.

Table 8: **Comparison with REACT on ImageNet for varying capacities.** Neural Priming outperforms REACT on ImageNet without adding parameters while retraining on 1% of the data REACT does (10 million vs 100,000.)

|  | B/16 | L/14 |
|---|---|---|
| REACT | 67.5 | 76.4 |
| Neural Priming | **68.38** | **78.72** |

original URL to get the image, which adds another factor of variation). Given the massive changes in performance between the two options for **(2)**, we only give wall-clock timing estimates for **(1)**.

FTS databases in SQLite use an inverted index over the tokens of their entries. In this case, the tokenization over full words and some more common suffix strings. Given this tokenization, it is easier to search for less common phrases and words. In Figure 7, we analyze this effect with respect to ImageNet priming pool construction. We see that most queries take less than 1 second to search all of LAION-2b. A few very common queries take much longer. For example, the most common query, T-shirt, takes >100s to search. The overall time to complete these searches for ImageNet was 21 minutes. For smaller datasets with less common class names, this process is significantly shorter. This is the majority of the runtime of our overall method.

## G.2  Exact Retrieval Time

The next part of our process was performing exact retrieval on the priming pool at test time. The method for this is described in Section 3.1.2 This is generally efficient, as the priming pool is a significantly filtered version of the pre-training dataset. To filter a priming pool of 1.1m images to 51k using the test set of ImageNetV2 took **3.2 minutes** on consumer hardware (two 3090 GPUs), not including the initial feature extraction time since these are easily pre-computed.

## H  Other Implementation Details

**Retrieval + Finetuning**  We retrieve 100 images per class using HNSW, an approximate nearest neighbors method, with the CLIP text-embedding from the class name. We use contrastive fine-tuning with the CLIP objective as the pretrain data only has natural language descriptions as labels. We fine-tune for with a learning rate of 1e-6 and batch size 1024.

**CoOp [55]**  CoOp tries to learn the embeddings for the prompts using a few examples for each of the classes. We use the official implementation[2] provided by the authors with default hyperparameters. For ImageNet experiments, we the models for 50 epochs regardless of the shots. For the smaller datasets like Stanford Cars, Flower102 and FGCV-Aircraft, we train the 1-shot models for 50 epochs, 2-shot and 5-shot models for 100 epochs, and 10-shot models for 200 epochs. We use the standard train-test split for all the datasets to ensure consistency across all the methods. Lastly, instead of the CLIP models provided by OpenAI, we use OpenCLIP [21] models for fair comparison against Neural Priming.

---

[2]https://github.com/KaiyangZhou/CoOp

Table 9: **Performance on Neural Priming (NP) with prompt-tuning method in the few-shot setting.** We find that prompt-tuning methods are compatible with Neural Priming even when training examples are available.

| Shots | ImageNet | | Stanford Cars | | Flowers102 | | FGVCAircraft | |
|---|---|---|---|---|---|---|---|---|
| | NP + CuPL | NP | NP + CuPL | NP | NP + CuPL | NP | NP + CuPL | NP |
| 0 | 71.38 | 71.15 | 90.23 | 90.11 | 80.04 | 79.78 | 35.70 | 35.38 |
| 1 | 71.56 | 71.32 | 90.31 | 90.17 | 80.77 | 80.72 | 36.39 | 36.55 |
| 2 | 71.81 | 71.50 | 90.55 | 90.21 | 88.90 | 88.60 | 38.16 | 37.94 |
| 5 | 72.03 | 71.92 | 91.03 | 90.93 | 94.66 | 93.43 | 40.01 | 39.84 |
| 10 | 72.36 | 72.27 | 91.54 | 91.49 | 94.83 | 94.75 | 43.83 | 43.67 |

**TPT [45]** Test-time Prompt Tuning (TPT) combines the ideas of test-time training [48] and prompt learning [55]. For a given test image, TPT tries to learn a soft prompt that maximizes agreement between multiple augmented views. We use the official implementation[3] for all the experiments and train TPT with 64 augmented views to ensure the best performance on all the datasets.

| $\sigma$ | ImageNet | Stanford Cars | FGVC-Aircraft | Flowers102 |
|---|---|---|---|---|
| 1 | 68.9 | 87.54 | 25.9 | 71.94 |
| 10 | 69.1 | 88.98 | 29.75 | 74.97 |
| 100 | 70.75 | 89.3 | 33.03 | 79.81 |
| 1000 | 70.64 | 89.11 | 31.32 | 78.81 |

Table 10: An ablation over the sigma value used to calculate the mixing coefficient between the image and text features. A larger sigma value is associated with a stronger prior for the text features.

# I   Confidence Intervals

We report the confidence intervals and average accuracy for the few-shot experiments (Table 11). For the zero-shot experiments, performance does not vary. For the few-shot experiments variation derives from sampling the training examples. We average accuracy across 10 runs and find the variance to be minimal (less than 1% for most settings).

# J   Broader Impact

One potential risk of Neural Priming is amplifying bias in the dataset. By retraining on a subset of the data, it is possible to condition the model in a way that is biased given a biased prompt or captions. For downstream tasks which are sensitive to bias, the risk could be mitigated by filtering the captions for inappropriate text or using an auxiliary model to detect inappropriate images. This approach of filtering the priming dataset is a potential direction for reducing bias in a model on the fly. Another potential drawback of Neural Priming is that it increases the carbon footprint of the trained model by retraining it on more data. However, the size of our priming set is small relative to LAION-2B (less than 1%) and we only retrain on the last layer which minimizes compute cost.

---

[3]https://github.com/azshue/TPT

Table 11: **Few-shot results with confidence intervals.** We report the average performance of Neural Priming across 10 runs with the standard deviation. We find that Neural Priming has lower variance than the baseline as the constant priming pool leads to less variation in the centroid of images.

| Shots | ImageNet | | Stanford Cars | | Flowers102 | | FGVCAircraft | |
|---|---|---|---|---|---|---|---|---|
| | Neural Priming | Baseline (NCM) | Neural Priming | Baseline (NCM) | Neural Priming | Baseline (NCM) | Neural Priming | Baseline (NCM) |
| 1 | $71.53 \pm 0.04$ | $69.81 \pm 0.05$ | $90.31 \pm 0.16$ | $88.57 \pm 0.48$ | $80.77 \pm 0.14$ | $76.09 \pm 1.14$ | $36.39 \pm 0.22$ | $27.56 \pm 1.12$ |
| 2 | $71.83 \pm 0.05$ | $70.05 \pm 0.05$ | $90.55 \pm 0.15$ | $88.73 \pm 0.43$ | $88.90 \pm 0.13$ | $82.50 \pm 0.74$ | $38.16 \pm 0.17$ | $30.01 \pm 0.67$ |
| 5 | $72.02 \pm 0.04$ | $70.43 \pm 0.02$ | $91.03 \pm 0.12$ | $89.19 \pm 0.18$ | $94.66 \pm 0.10$ | $89.46 \pm 0.29$ | $40.01 \pm 0.11$ | $35.50 \pm 0.77$ |
| 10 | $72.34 \pm 0.03$ | $70.90 \pm 0.04$ | $91.54 \pm 0.10$ | $90.86 \pm 0.16$ | $94.83 \pm 0.01$ | $92.74 \pm 0.02$ | $43.83 \pm 0.09$ | $40.84 \pm 0.42$ |

