# OpenReview forum: "Neural Priming for Sample-Efficient Adaptation"
_NeurIPS.cc/2023/Conference — NeurIPS 2023 poster_

### Official Review · Reviewer_AXQG · 2023-07-06

**Soundness:** 4 excellent
**Presentation:** 3 good
**Contribution:** 3 good
**Rating:** 7
**Confidence:** 4

**Summary:**

This paper introduces neural priming, a retrieval based approach for adapting large pretrained models to downstream tasks. Neural priming first collects a priming pool of pretraining data relevant to the downstream task using natural language descriptions, then at test-time the priming pool is further filtered by keeping the top-k most similar pretraining samples to the test samples w.r.t. cosine similarity of the CLIP embeddings. From this filtered pool, a classification head is set to the nearest-class mean and averaged with the CLIP ZS text embeddings (similar to WiSE). Across several classification datasets, neural priming is shown to improve over existing prompt-tuning and test time adaptation methods in the zero-shot and few-shot settings, and is complementary to prompt-tuning methods.

**Strengths:**

The problem of fitting pretrained classifiers to downstream tasks effectively, specifically models like CLIP, is of great interest to the ML community and the retrieval based approach of neural priming is both simple and a pleasant departure from current robust finetuning methods. Overall the paper was well-written, easy to follow, and the evaluation shows consistent improvements over existing prompt tuning and retrieval-based transfer learning methods. I also enjoyed the open questions posed in the discussion section on why retrieval methods work well for transfer learning which I could see sparking future work in this area.

While the method is simple and there are some eval limitations that I would like addressed, I think this paper is of value to the community and is above the acceptance threshold.

**Weaknesses:**

Given the simplicity of the method and the vast number of related works, my critiques mainly lie in differentiating from related works more and adding in some simple yet relevant baselines.

**Related works**

My main critique with the related works section is the lack of differentiation from test-time training/adaptation and transductive learning in section 2.3, as well as few-shot learning in section 2.4.

Not a critique and more of a suggestion, but a recent paper[4] explores how removing the least relevant items in a pretraining dataset can improve transfer learning accuracy and I believe would be good motivation for neural priming. There are also some robust CLIP finetuning methods[1] and prompt tuning methods[2,3] that should be mentioned in related works.

[1] [Using Language to Extend to Unseen Domains](https://arxiv.org/abs/2210.09520) \
[2] [Visual Prompt Tuning for Test-time Domain Adaptation](https://arxiv.org/abs/2210.04831) \
[3] [Conditional Prompt Learning for Vision-Language Models](https://arxiv.org/abs/2203.05557) \
[4] [A Data-Based Perspective on Transfer Learning](https://arxiv.org/abs/2207.05739)

**Evaluation**
Here the main thing that is lacking is a description of the baselines. There is some brief descriptions of CoOp (without a citation) but I found it hard to interoperate exactly what these baselines were. \
Why not compare to CLIP LP or WiSE for the few shot setting? It seems like that would be a natural baseline (even just setting the ZS weights to the few shot examples, from what I understand the NCM baseline uses the priming examples).

**Questions:**

Does the CLIP ZS baseline in Table 1 just include the class name or does it also include context (i.e. 'a photo of a {} flower' instead of 'a photo of a {}'")?

**Limitations:**

One big limitation that I did not see mentioned was the time it takes to retrieve the priming pool from the original pretraining data, especially for large datasets like LAION 2B.

---

> ### Author Rebuttal · Authors · 2023-08-09
>
> Thanks for your thorough and thoughtful review. We appreciate your insightful feedback, and hope to address your concerns below.
>
> **Given the simplicity of the method and the vast number of related works, my critiques mainly lie in differentiating from related works more and adding in some simple yet relevant baselines.**
>
> Thanks for the feedback. We recognize that the discussion in related work could have provided better context for Neural Priming. We have added further related works and added discussion about the differences with Neural Priming.
>
> To our knowledge, the most closely related work is REACT [1]. This paper retrieves images from an external data source, and fine-tunes with a gating mechanism. First, Neural Priming differs in that we keep the base architecture fixed whereas REACT adds additional parameters. Second, Neural Priming shows that performance can be improved by finding subsets of the original pretraining data, whereas REACT [1] and previous methods [2,3] add data from external sources. Third, Neural Priming is magnitudes more efficient and can be on-the-fly for transductive learning and domain-shift.
>
> For few-shot works we’ve added the following works with discussion based on feedback from reviewer 89m2 and further literature review.
> - Atlas: Few-shot Learning with Retrieval Augmented Language Models,
> - Training-free name-only transfer of vision-language models
> - Few-Shot Learning Through an Information Retrieval Lens
>
> For transductive learning we’ve added the following with discussion:
> - Re-ranking for image retrieval and transductive few-shot classification.
> - ZegCLIP: Towards Adapting CLIP for Zero-Shot Semantic Segmentation
>
> **Related works section lacks of differentiation from test-time training/adaptation and transductive learning, as well as few-shot learning.**
>
> We realize the difference between test-time training and transductive learning was not well delineated in the related works. We’ve better explained that test-time training is online and typically handles one test example at a time whereas transductive has access to the entire test set. We’ve also added discussion to make it clear that few-shot differs in that the model has access to a few labeled training examples and no access to the test set.
>
> **The main thing that is lacking is a description of the baselines. There is some brief descriptions of CoOp (without a citation) but I found it hard to interpret.**
>
> Thanks for the feedback, we’ve added the citation for CoOp [4] in the experiments section and added details for the NCM and CoOP baseline. CoOp is a popular few-shot method for language-vision models which learns context vectors for each class given a set of labeled examples. In other words, rather than hand-engineering templates such as "a photo of {}, a type of flower", CoOp learns the templates from few-shot data. Based on your later comment, we’ve also added WiSE-FT as a baseline and clarified that NCM uses the few-shot examples with weight ensembling. We elaborate further on the details of NCM below and have added them to the experiments section.
>
> **Questions:**
>
> **1. Does the CLIP ZS baseline in Table 1 just include the class name or does it also include context (i.e. 'a photo of a {} flower')?**
>
> Yes, we’ve used the openAI templates such as “a photo of {}, a type of flower”, “a photo of {}, a type of aircraft”, etc. We’ve added this detail to the experiments section.
>
> **Limitations:**
>
> **1. One big limitation was the time it takes to retrieve the priming pool from the original pretraining data.**
>
> One of the major advantages of Neural Priming is the speed compared to similar methods. In figure 7 of the appendix and figure 2 of the rebuttal pdf we plot the wall clock time for retrieving the 1000 classes of ImageNet from LAION-2B. In total to retrieve 1 million images from LAION-2B it takes ~10 minutes on one rtx-6k and evo 970 SSD. For comparison, to retrieve the 10 million images required for REACT [1] using approximate nearest neighbors (FAISS) with text-to-image similarity on the same hardware takes ~72 hours.
>
>
> **2. Why not compare to CLIP LP or WiSE? From what I understand the NCM baseline uses the priming examples.**
>
> To clarify, the NCM baseline uses the few-shot training examples, not the priming examples. Also NCM uses the same weight interpolation between the centroid of the few-shot examples and the language classifier as Neural Priming. We also performed standard fine-tuning with weight ensembling and found that NCM with weight ensembling performed better, which has been observed for few-shot settings [5,6]. We realize this wasn’t clear, and have added details to the experiments section. We’ve also added the WiSE-FT baseline to the paper and the results on ImageNet can be found below.
>
> Top-1 Few-Shot Accuracy on ImageNet
>
> | Shots | Neural Priming | NCM  | Wise-FT |
> |-|:-:|:-:|:-:|
> | 0     |     71.38      | 68.3 |  67.75  |
> | 1     |     71.56      | 68.89|  68.14  |
> | 2     |     71.81      | 68.85|  68.25  |
> | 5     |     72.03      | 69.12|  68.78  |
> | 10    |     72.36      | 70.12|  69.02  |
>
> [1] Learning Customized Visual Models with Retrieval-Augmented Knowledge
>
> [2] K-LITE: Learning Transferable Visual Models with External Knowledge
>
> [3] Internet Explorer: Targeted Representation Learning on the Open Web
>
> [4] Learning to Prompt for Vision-Language Models
>
> [5] Prototypical Networks for few-shot learning
>
> [6] Rethinking Few-Shot Image Classification: a Good Embedding Is All You Need?

---

> > ### Comment · Reviewer_AXQG · 2023-08-12
> > **Update**
> >
> > I want to thank the authors for clearing up my concerns related to the baselines and differentiating from prior work, and believe that with these clarifications this paper provides an effective and easy to implement retrieval method. Therefore, I have raised my score from a weak accept to an accept.

---

> > > ### Author Response · Authors · 2023-08-16
> > > **Thanks For the Update**
> > >
> > > Thank you for updating us. We are glad that our rebuttal resolved the concerns you had. We appreciate the valuable feedback and suggestions you've provided which we think has significantly improved the clarity of the paper.

---

### Official Review · Reviewer_Ekvm · 2023-07-08

**Soundness:** 3 good
**Presentation:** 3 good
**Contribution:** 2 fair
**Rating:** 7
**Confidence:** 4

**Summary:**

The paper proposes a technique for adapting large pre-trained models to distribution shifts and downstream tasks given few or no labeled examples named Neural Priming. The method can be used at test time on very large scale datasets. Experiments on various datasets show the effectiveness of Neural Priming.

**Strengths:**

1. The proposed Neural Scaling method is novel, with a good motivation (diversity of pretraining datasets introduces competing objectives) and clear presentation.
2. The technical details of Neural Scaling is simple yet effective: Classical ideas of information retrieval, clustering and prototype calculation were used. This may possibly open a line of research on advancing the proposed techniques.
3. Neural Scaling has high efficiency on saving the time of doing linear probing.

**Weaknesses:**

1. In line 182, what is P and \sigma? The main text should include their meanings rather than putting them in the appendix. Also, ablation studies/discussions can be included for the choice of \sigma.
2. I am not familiar with the performance of the compared baselines e.g. VLM (whether these methods are strong and competitive), thus I cannot determine to which extent Neural Priming outperforms other relative methods, especially SOTA ones. While this do not effect the paper's advantage in novelty, it does draw questions on the "absolute" performance of Neural Priming.

**Questions:**

The questions here are for open discussion.
1. I wonder if all the filtered priming pools for a specific downstream task are integrated into one dataset, will it be a better pretraining/finetuning dataset (excluding noises)?
2. Is there a possibility that some samples are harmful (e.g. low quality or wrongly labeled), but selected into the priming pools or even the filtered priming pool?

**Limitations:**

Limitations are discussed in terms of the method itself. But issues on broader aspects such as model safety, data leakage are not included.

---

> ### Author Rebuttal · Authors · 2023-08-09
>
> Thanks for your thorough and thoughtful review. We hope to address your concerns below and are happy to engage in further discussion.
>
> **Weaknesses:**
>
> **1. In line 182, what is P and $\sigma?$ The main text should include their meanings rather than putting them in the appendix. Also, ablation studies/discussions can be included for the choice of $\sigma.$**
>
> P is the size of the priming pool, and we’ve made that more clear in the text. The formula for the mixing coefficient, $\alpha$ can be derived by considering the language embedding as a Gaussian prior and updating the prior with examples from the priming pool. $\sigma$ is the certainty over the language embedding prior. We’ve included an ablation over the choice of $\sigma$ below.
>
> | $\sigma$ | ImageNet | Stanford Cars | FGVC-Aircraft | Flowers102 |
> |:-:|:-:|:-:|:-:|:-:|
> | 1      | 68.9     | 87.54         | 25.9          | 71.94      |
> | 10     | 69.1     | 88.98         | 29.75         | 74.97      |
> | 100    | 70.75    | 89.3          | 33.03         | 79.81      |
> | 1000   | 70.64    | 89.11         | 31.32         | 78.81      |
>
>
>
>
>
>
>
>
> **2. I am not familiar with the performance of the compared baselines e.g. VLM (whether these methods are strong and competitive), thus I cannot determine to which extent Neural Priming outperforms other relative methods, especially SOTA ones. While this do not effect the paper's advantage in novelty, it does draw questions on the "absolute" performance of Neural Priming.**
>
> VLM [1]  is a recent paper published at ICLR 2023 which to our knowledge is state of the art for zero-shot along with CuPL [2]. Based on the recommendation from Reviewer XXJ2 we’ve added the TPT+CoOp and TENT [1] baseline for the transductive and few-shot setting which can be found below. If there are other methods that should be considered we’d be happy to compare with them.
>
> Top-1 Accuracy on Few-Shot ImageNet
> |             | 1-Shot  | 2-Shot  | 5-Shot  | 10-Shot |
> |:-:|:-:|:-:|:-:|:-:|
> | TPT+CoOp       | 66.97   | 70.04   | 71.02   | 71.89   |
> | Neural Priming | **71.56** | **71.81** | **72.03** | **72.36** |
>
> Comparing with TENT in Transductive Setting
>
> || ImageNet-V2| ImageNet-Sketch |
> |:-:|:-:|:-:|
> | CLIP| 59.35|57.05 |
> | TENT|59.91|57.40|
> | Transduct. Priming |60.76| 59.97|
>
> **Questions:**
>
> **1. I wonder if all the filtered priming pools for a specific downstream task are integrated into one dataset, will it be a better pretraining/finetuning dataset (excluding noises)?**
>
> If we understand correctly, the question is if we take all of the priming pools across downstream tasks and combine them, will it boost accuracy in general. That's an interesting idea. With the 11 tasks we evaluate in the paper, the diversity of classes is likely too small for generalizing to new tasks (i.e. most classes for a new task would not be in our combined priming pool). If hundreds or thousands of tasks were used to create a combined priming pool, it's quite possible that it could produce a fine-tuning dataset that would be more accurate for new tasks.
>
>
> **2. Is there a possibility that some samples are harmful (e.g. low quality or wrongly labeled), but selected into the priming pools or even the filtered priming pool?**
>
> Yes, during our experiments we found that using the priming pool before filtering with CLIP similarity score (zero-shot setting) or image similarity (transductive setting) can decrease accuracy.
>
>
> **Limitations:**
>
> **1. Limitations are discussed in terms of the method itself. But issues on broader aspects such as model safety, data leakage are not included.**
>
> We agree and have added content on model safety to the limitations section. We recognize that biased class names could lead to harmful images being retrieved to the priming pool and lead to unintended behavior of the model. We also agree that data leakage is an issue with large-scale models pretrained on data scraped from the internet. We suspected there might have been data leakage so we ran deduplication between the images in the priming pool and images in the test sets, but found no overlap. We contend that data leakage is more of an issue with the pretraining set since Neural Priming does not introduce any new data.
>
> [1] Tent: Fully Test-Time Adaptation by Entropy Minimization

---

> > ### Comment · Reviewer_Ekvm · 2023-08-13
> > **Thank you for your response**
> >
> > The authors have resolved my questions. I am raising my score.

---

> > > ### Author Response · Authors · 2023-08-16
> > > **Thanks Reviewer Ekvm**
> > >
> > > Thanks for letting us know. We are glad that our rebuttal resolved your questions and appreciate your constructive and valuable feedback.

---

### Official Review · Reviewer_89m2 · 2023-07-11

**Soundness:** 3 good
**Presentation:** 4 excellent
**Contribution:** 2 fair
**Rating:** 4
**Confidence:** 5

**Summary:**

This paper presents a novel approach to zero-shot learning in vision-language models. The authors propose a new tunning schema using prior knowledge in the CLIP model. First, the authors construct a priming pool, which consists of clusters of retrieval images by the CLIP model. Each cluster collects a specific class by searching all text-image pair owning that class name in the text. The authors then tuned this priming pool using the test set. For each image in the test set, the 10 best matching images are selected and de-duplicated to make up the final priming pool. The authors claim that such an operation can adapt the priming pool to the test domain. At last, the authors tune the CLIP model by adding a linear head at the end of the model. The linear head is the weighted sum of a zero-shot trained head and NCM of all retrieved examples per category. Experiments show a 2.45% improvement on ImageNet and a 3.81% improvement on average across some traditional transfer learning benchmarks. Overall, this paper provides valuable insights and promising directions for future research in this field.

**Strengths:**

# Novelty:
This paper provides a novel perspective of aligning images by the class name directly. Previous methods are often trapped in playing with feature representations and embeddings. None has looked into whether using strings directly, is a good practice. This work fills this unseen point with good improvements. The method can leverage massive training data, to help the model adapt better to specific tasks.

# Clarity:
This paper is extremely well-written and easy to follow.

**Weaknesses:**

# Novelty:
The key innovation of this work can be split into three parts, 1) **class name retrieval to construct priming pooling**, 2) **filtering the priming pooling using cosine similarity with the test set**, and 3) **tuning the CLIP model by adding a linear head using NCM of the priming pooling**. So I will give my opinion to them separately as follows.

1. **Retrieval**:

a) **Retrieval is not a new idea in transfer and zero-shot learning.** I think using retrieval is not new in both domains of transfer learning or few-shot\zero-shot learning, like in NeurIPS2021,2017 paper "Re-rankingforimageretrievalandtransductive few-shot classification", "Few-ShotLearningThroughanInformation RetrievalLens", there may be too many to list.

b) **String retrieval  I think is not powerful enough.** So the new stuff is using the class name as the retrieval feature. This is somewhat too direct for me. Is there a possibility that CLIP representations combining the string retrieval perform better? According to my experience, this is usually the case. The authors do not discuss the possibilities of using other tools, instead, they come out with a belief that string retrieval is the fastest one so it is superior to other methods, which I may respectfully disagree. First, time efficiency should not be a major problem in constructing the priming pool, after all, it is not required to be online learned or real-time computed. In your pipeline, you first get the naive priming pool from the training data. Then you prune it to preserve the top 10 similar images with each image of the test data. This step won't consume more time if you use semantics as you need to go through all the data in the priming set either way. Then at inference time you only need to compute the mean of each class, there is no need for exact string matching so it also won't consume more time if you use semantics or other stuff. So using semantics or not won't influence the inference time at all, I can't understand why only string is used. Perhaps some experiments comparing those retrieval methods can be helpful to demonstrate the advantages of this work.

c) **Potential conflicts.** Another problem of only using string retrieval is that it only conflicts with each other. Like the class "guinea pig" is not a pig at all, "wolf spider" is not a wolf, and "sea snake" may not contain sea in the image. I am interested in how this method can tell the "guinea pig" category (339 in ImageNet) and the "pig" category (342 in ImageNet)? On the hand, combining string with semantics can handle this case, as the embedding of guinea pigs will be more similar to the mouse's. Also, I don't understand why the results of string retrieval will be more relevant, can the authors explain why?

2. **Filtering the priming set.** I think the authors can try combining CLIP similarity with Pixel similarity in ranking, inducing more metrics to cooperate. In transfer learning and few-shot learning, such methodology is usually adopted.

3. **NCM head.**  This step uses the final outputs of the language model to predict the mean. Why not also use the output of the Image encoder? Or how about the intermediate representations of the CLIP model? I think the authors can add an explanation of why the output of the language model is the best. Also, tuning a new head is not a new thing. It is popular in incremental learning, transfer, and zero-shot learning. But it is not necessary to limit your imagination in here. Many works have discovered some other parts of the CLIP model, like attention layers, and convolutional kernels, MLPs have rich semantics and an important role in making predictions. If the authors can discuss why they choose to add a linear head instead of tuning other parts, I think this section can be more convincing.  For example, the authors can compute the gradients of each parameter when input new examples of test sets, and find out that the terminal linear layer has the most significant gradients thus influencing the results on new examples the most, so tuning them is vital and more efficient than tuning other parts.

Overall, the three parts are a bit split and there seem no strong connections among them, at least not provided by the authors. They seem like A+B+C. If the authors can reorganize them, and engage them together with one purpose of enhancing transfer learning capability, I think this paper can be more fluent.


# Experiments:
1. **No errors and error bars.** While it is recommended by NeurIPS author guidelines, papers in this domain seem seldom report error bars. I think it could enhance the paper as some figures are close, and errors can help us understand the true advantage of the proposed method. As in Tab. 1, in ImageNet, FGVC, Food101, and SUN397, the differences are smaller than 0.5. We may suspect it within the variance of measures. Also in Fig.3, performance is close, error bars can further tell the differences between methods.
2. **Why no LAION-5B?** I see many relevant papers conducted on the LAION-5B dataset. LAION-2B is a new one and includes it very well. However, omitting LAION-5B may be hard to align the results of this paper to previous work.
3. **Ablation is not enough.** Like in the novelty part, the superiority of the chosen components to their alternatives need to be clarified by the experiments. This will make this submission stronger.
4. ""Fair comparason.** As you add a head to the model, you may control the equivalence of all parameters and training steps of all methods. This is important when performance is close.



# Minor issue:
1. Line 143 "However,143 withneuralpriming,prioritizingprecisionoverrecalliscrucial,consideringthesize,diver", better to add references.
2. Line 164, the last "for" is redundant.
3. It is not very clear whether Neural Priming is another name of this method or the task this method tries to handle. Better clarify and unify it throughout the paper.

**Questions:**

1. Will the method improve finetuning efficiency of other prompt tuning methods? I mean after applying it, how many training times or steps can be saved for prompt-tuning?
2. Is the method stable among different tasks?
3. How is the influence of the filtering step? If remove it, how much accuracy will lose? Also, is there any ablation on the choice of $\alpha$?



The following are suggestions rather than comments, the authors may take them advisedly.
1. **Notations.** I am in favor of notations used in SimCLR, which are much clear to read and easy to follow as they use different font styles for different concepts. I would suggest a slightly modified version as follows: sets, using \mathcal command like $\mathcal{S}$; neural networks, using bold font with parameter subscript, like $f_{\theta}$, number field, like the real number set, using $\mathbb{R}^d$, integers using $\mathbb{N}$; loss function, using \mathcal command with subscript the \mathrm environment, like $\mathcal{L}_{\mathrm{CLIP}}$; scalars using plain font and vectors using bold fond to distinguish. Currently, the notations carry only limited meaning, make readers need to look back to their definitions several times to remember them. For example, Line 130 uses $\mathcal{R}^d$ for real numbers while Line 178 uses $\mathbb{R}^{d\times n}$, V,P,B,L stand for model, data pool (while training data using font $\mathcal{D}$), data clusters, and model but using the same font style.  I also recommend not to use L as a notation for the model, it can be confused with the loss function.

2. **Fontsize of figures.** Currently fontsizes of figures are set carelessly. Fig. 1 using a giant fontsize, much larger than the context, seems not necessary, while some words in Fig.2 are too small to tell. I also recommend you to use some images to tell readers the meaning of each component in Fig. 1, like putting some training images above the pretraining data, and putting some images from the priming pool beneze the priming set, using a neural network model to replace the CLIP block and color those parameters that need to be tuned.

3. **The hyphen.** In the title you use **Data-Efficient**, words connected by the hyphen are in capital. While in **Zero-shot**, **Nearest-class-Mean**,  they are not.

**Limitations:**

1. The method needs to know all test set to filter the priming set, online learning may exceed the scope of it.
2. Filtering step may consume considerable time and computation resources, not affordable for ordinary researchers.

---

> ### Author Rebuttal · Authors · 2023-08-09
>
> Thank you for the thorough review. We hope our response addresses your concerns. Due to the character limit during the rebuttal phase, we’ve focused on the most important concerns and will address the remaining comments at the start of the discussion period.
>
> **Using retrieval is not new in domains of transfer, few-shot, and zero-shot learning like in [1], [2].**
>
> Thanks for pointing us to these works. We have added them to our paper with discussion. The use of retrieval in general is not new, but these works differ significantly from Neural Priming in both overall objective and method details. The goal of our work is to efficiently improve vision-language models by leveraging their large-scale pretrain dataset. [1] formulates a new optimization based on information retrieval for training small convolutional models from scratch. Also, the method requires human annotated labels, whereas Neural Priming can leverage web captioned data. [2] formulates a training objective for learning a better similarity graph. Similar to [1], [2] uses human annotated images and is focused on better pretraining from small-scale datasets.
>
> **String retrieval is not powerful enough. Authors should try combining CLIP similarity with Pixel similarity in ranking.**
>
> Yes, that is exactly what Neural Priming does (line 156-159). We do a multi-stage filtering and string retrieval is just the first phase. After the initial filtering phase, the more expensive techniques of CLIP similarity or image embedding similarity (transductive setting) are used.
>
> **Time efficiency should not be a problem in constructing the priming pool.**
>
> Agreed, time-efficiency is not a major concern for the priming pool. However, performing exact nearest neighbor search over large-scale datasets such as LAION-2B is infeasible on current hardware. Other works [3] use approximate nearest neighbors, but this leads to empirically worse results as can be seen from the retrieval ablation study in Table 1 of this response and in Table 1 of the main paper.
>
> **Experiments comparing retrieval methods can demonstrate the advantages of this work.**
>
> The baseline of retrieval+fine-tuning uses approximate nearest neighbors search. We found that the approximate semantic search retrieved many images that did not fall into the target class. Below we’ve included a table showing the comparison of different retrieval methods and have added it to the paper.
>
> Table 1. Ablation Over Filtering Methods
> || ImageNet | Stanford Cars | FGVC-Aircraft | Flowers102 |
> |:-:|:-:|:-:|:-:|:-:|
> | CLIP| 68.3| 87.4|25.86| 71.65|
> | Retrieval + Fine-Tune|70.28|87.95|26.22|72.15|
> | Neural Priming (No CLIP Filter)| 70.35| 88.19| 31.56|76.91|
> | Neural Priming (No Label Alignment)|70.64| 88.38| 30.38|77.18|
> | Neural Priming| **70.75**| **89.3**| **33.03**| **79.81**|
>
> **Using string retrieval can cause conflicts.The class "guinea pig" is not a pig at all, "wolf spider" is not a wolf".**
>
> This is a good observation and a major reason we introduced CLIP similarity to further filter these images (section 3.1.1). In Figure 1 of our rebuttal PDF, we show random examples of images removed by the CLIP similarity filter for the "pig" class and the remaining "pig" images in the priming pool. From Figure 1, the CLIP filtering step removes essentially all of the guinea pigs. We will add this motivation and figure to the paper.
>
> **NCM uses the outputs of the language model, why not also use the output of the Image encoder?**
>
> We do use the output of the image encoder to construct the centroid for NCM.  On line 175-176: *we perform nearest-class mean (NCM) on all retrieved examples per category to get a pool specific classification head*.
>
> **No errors and error bars. The differences are smaller than 0.5. Fig. 3 has no error bars.**
>
> For Table 1, zero-shot and transductive results do not have error bars because zero-shot inference is deterministic. Also, the improvement on ImageNet over CLIP is 3.08% which is considerable for this benchmark.
>
> We provide confidence intervals (CI) for the few-shot experiments in Table 8 (section I of the appendix). The CI for ImageNet is .05% and the average improvement across all shot numbers is 1.63%. For Figure 3, the error in table 8 is so minimal that it is not visible in the plot. Average improvement is 3.57% across the 4 datasets listed, and average CI is .103%.
>
> **The superiority of the chosen components to their alternatives need to be clarified.**
>
> In Table 1 of this response we’ve presented an ablation study over the various retrieval components in Neural Priming which we’ve added to the paper.
>
> **As you add a head to the model, you may control the equivalence of all parameters and training steps of all methods.**
>
> All methods we benchmark against require a linear head, therefore Neural Priming has the same or fewer new parameters. Our method does not use gradient steps and is ~155x more efficient on average in the number of training FLOPS than REACT [3], TPT [4], and CoOp [5].
>
> **Filtering step is not computationally affordable for ordinary researchers.**
>
> We agree that accessibility and computational efficiency are important. One motivation for Neural Priming is to enable others to leverage large-scale data sources without needing massive compute budgets. Neural Priming can be run on a single machine with a 3090 TI and 1 TB SSD. Comparable methods such as REACT [3] require retrieving and fine-tuning on 10 million auxiliary images (100,000 times more images than Neural Priming) which we agree can be prohibitively expensive.
>
> [1] Few-Shot Learning Through an Information Retrieval Lens
>
> [2] Re-ranking for image retrieval and transductive few-shot classification
>
> [3] Learning Customized Visual Models with Retrieval-Augmented Knowledge
>
> [4] Test-Time Prompt Tuning for Zero-Shot Generalization in Vision-Language Models
>
> [5] Learning to Prompt for Vision-Language Models

---

> > ### Author Response · Authors · 2023-08-11
> > **Response to Additional Reviewer Comments**
> >
> > We appreciate your thorough review. Below we've responded to the remaining review comments that did not fit into our initial rebuttal. We are happy to engage in further discussion and are interested in any further comments you may have.
> >
> > **Ablation on the choice of alpha?**
> >
> > We’ve included an ablation on the choice of $\sigma$ which determines the value of $\alpha$ according to $\alpha = e^{-|P|^2/\sigma}$ where $|P|$ is the cardinality of the priming pool. This formulation can be viewed as a Gaussian prior over the language embedding with posterior updates from the priming pool. Below we've included a table with an ablation over $\sigma$ and added these details to the paper.
> >
> > Ablation over choice of $\sigma$. Zero-shot top-1 accuracy is reported on various downstream tasks.
> > | $\sigma$ | ImageNet | Stanford Cars | FGVC-Aircraft | Flowers102 |
> > |--------|----------|---------------|---------------|------------|
> > | 1      | 68.9     | 87.54         | 25.9          | 71.94      |
> > | 10     | 69.1     | 88.98         | 29.75         | 74.97      |
> > | 100    | 70.75    | 89.3          | 33.03         | 79.81      |
> > | 1000   | 70.64    | 89.11         | 31.32         | 78.81      |
> >
> > **Questions:**
> >
> > **Will the method improve fine tuning efficiency of other prompt tuning methods? I mean after applying it, how many training times or steps can be saved for prompt-tuning?**
> >
> > Neural Priming does not improve training efficiency compared to the base CLIP model. The method improves sample-efficiency. In other words, a neural primed model can achieve the same accuracy as the base model with fewer labeled examples from the target distribution. The Neural Priming does improve training efficiency compared to other retrieval augmented methods such as REACT [7].
> >
> > **Why not use LAION-5B?**
> >
> > LAION-2B is the english subset of LAION-5B commonly used by other works for pre-training and benchmarking[6,7]. For instance, OpenCLIP models are trained on LAION-2B and not LAION-5B. LAION-5B is comprised of LAION-2B, LAION-multilingual, and LAION aesthetics. We will add this clarification to the paper.
> >
> > **Is the method stable among different tasks?**
> > Across all of the datasets we benchmarked, Neural Priming provided consistent accuracy improvements. If the question is about training stability, our method is deterministic and therefore stable from an optimization perspective. If the question is whether Neural Priming works for tasks such as segmentation, that is an interesting future direction.
> >
> > **Limitations:**
> >
> > **The method needs to know all test set to filter the priming set, online learning may exceed the scope of it.**
> >
> > Fair point, though Neural Priming would likely work in online settings. From our few-shot experiments, the transductive filtering approach works even with only a few samples from the target distribution.
> >
> > **The following are suggestions rather than comments, the authors may take them advisedly.**
> >
> > Thanks for the suggestions regarding notation, font size of figures, and use of capitalization with hyphens. We have updated the paper accordingly to reflect your comments and improve the readability of the work.

---

### Official Review · Reviewer_xxj2 · 2023-07-28

**Soundness:** 3 good
**Presentation:** 3 good
**Contribution:** 3 good
**Rating:** 6
**Confidence:** 4

**Summary:**

This work proposed a new method, named Neural Priming, which uses retrieval to adapt vision-language foundation models to downstream tasks given few or no labeled examples in the test time. Specifically, Neural Priming first uses the set of class names in a new task to retrieve relevant samples from the pre-training dataset to form the priming pool and also uses the test examples to narrow the priming pool. Then, the CLIP model can be attuned using the priming pool by constructing the linear prediction head. The experiments are performed in three settings: zero-shot, few-shot and transductive.

**Strengths:**

1. The idea of using retrieval to adapt large vision-language models to downstream tasks is interesting and intuitive. Compared with similar approaches, such as REACT, this work does not need to add new trainable modules.
2. Experiments on several settings (zero-shot, few-shot and transductive) show the effectiveness and broad applicability of the proposed method. Ablation studies on the number of samples per class in the priming pool and the model capacity are interesting.

**Weaknesses:**

1. Regarding the retrieval idea for test-time adaptation, the major difference between Neural Priming and REACT is that REACT adds new training modules while Neural Priming only constructs a linear prediction head. It is not clear to me how significant this difference is and how original the proposed method is. Also, there is no direct comparison between these two approaches in experiments. Since Neural Priming and REACT are used in similar settings, it is not clear to me if the proposed method can largely improve over REACT.
2. It is not clear how sensitive the proposed approach is to the retrieval database. From experiments, we see the improvement will be less significant if the target dataset is well presented in the pre-training dataset (see lines 237-241). On the other hand, the performance will also drop if there is a large gap between the target dataset and the retrieval dataset. Another question is whether the retrieval database has to be the pre-training dataset? Or we can arbitrarily construct a large-scale dataset as retrieval database.
3. In the few-shot setting, what is the comparison with CoOp+TPT (which is better than CoOp according to [44])?
4. In the transductive setting, it is unfair to compare TPT as TPT only relies on a single test sample while the proposed method takes all samples in the test set into account. I think another baseline called TENT (https://openreview.net/forum?id=uXl3bZLkr3c) should be considered.
5. Writing can be improved. There are many repeated sentences across the whole paper. For example, the quantitative results have been mentioned in both introduction and experiment sections repeatedly.
6. Not sure what “3-shot CLIP model” refers to in line 214.

**Questions:**

My major concerns and questions are in how the experiments support the effectiveness and significance of the proposed method. Please see the weaknesses part for more details. I’m willing to increase my rating if my concerns can be well addressed.

**Limitations:**

The authors have adequately addressed the limitations.

---

> ### Author Rebuttal · Authors · 2023-08-09
>
> Thanks for your thorough review. We are glad you found the paper interesting and intuitive and appreciate the feedback. We hope our response addresses your concerns and interested in any further comments you may have.
>
> **1. Regarding the retrieval idea for test-time adaptation, the major difference between Neural Priming and REACT is that REACT adds new training modules while Neural Priming only constructs a linear prediction head.**
>
> At a high level, Neural Priming has four main contributions when compared with REACT [1] and other methods [2]. The first is that Neural Priming is far more sample-efficient in the number of examples drawn from the pretraining set. REACT [1] fine-tunes with 1-10 million examples whereas Neural Priming uses as few as 10,000. Second, Neural Priming is far more efficient in the retrieval process due to the initial text filtering step. Neural Priming takes around ~10 minutes on one rtx-6k and a Samsung Evo 970 SSD, whereas REACT requires ~72 hours. Third, due to the multi-stage filtration and the label realignment phase (captions are changed to fit the target categories), Neural Priming achieves 2.32% better performance on ImageNet. Fourth, we do not introduce new parameters or have architecture-specific modules, making Neural Priming compatible with any open-vocabulary model. Finally, we show that for even a fixed dataset, our method can find subsets which improve downstream performance. Previous methods [1,2,3] are focused on improving performance by adding new data to the model. This is a useful scientific contribution.
>
> **There is no direct comparison between these two approaches in experiments.**
>
> In table 6 on page 15 of the appendix we included a comparison on ImageNet between REACT and Neural Priming. The REACT results are taken from [1] as reproducing their experiments would be prohibitively expensive. We’ve also included the numbers in the below table for your reference. We did not include direct comparison in the main paper for two reasons. REACT adds a significant number of new parameters to the model, and requires 1-10 million retrieved samples whereas Neural Sampling uses 10-100 thousand. Still, Neural Priming performs favorably.
>
> Top-1 Accuracy on Zero-Shot ImageNet
>
> || B/16  |L/14|
> |:-:|:-:|:-:|
> | REACT|67.5|76.4|
> | Neural Priming |**68.38**|**78.72**|
>
> We’d also like to note that the baseline, retrieval+fine tuning, in Table 1 of the main paper uses an implementation of REACT without adding parameters and using the same number of retrieved samples as Neural Priming for a direct comparison.
>
> **2. How sensitive is the proposed approach to the retrieval database? Does the retrieval database have to be the pre-training dataset?**
>
> We’ve run experiments using LAION-2B trained model to retrieve from the LAION-400M database which can be found below. We find similar results, albeit retrieving from smaller scale databases improves accuracy less. We primarily focused on using the pretraining dataset as the retrieval database to understand the effect of resampling data without adding new data. It’s clear from past work that adding more data will improve models and one of our contributions is showing that performance can be improved even with the data held fixed.
>
> Ablation over Retrieval Database.
> || ImageNet | StanfordCars | FGVC-Aircraft | Flowers102 |
> |:-:|:-:|:-:|:-:|:-:|
> | CLIP|68.3|87.4|25.86|71.65|
> | Priming (LAION-400M)| 70.21|88.14|30.52|76.23|
> | Priming (LAION-2B)  |70.75|89.3|33.03|79.81|
>
>
> **3. In the few-shot setting, what is the comparison with CoOp +TPT (which is better than CoOp according to [44])?**
>
> CoOp+TPT is not a straight-forward comparison in the few-shot setting as TPT requires training on the test sample which the other few-shot methods are not designed for. We’ve run the CoOp+TPT baseline and report the results below. Also CoOp+TPT takes 12 hours on one RTX-6k and is not feasible to run on-the-fly. Even with these caveats, we still outperform CoOp+TPT.
>
> Top-1 Accuracy on Few-Shot ImageNet
>
> || 1-Shot  | 2-Shot  |5-Shot  |10-Shot
> |:-:|:-:|:-:|:-:|:-:|
> | CoOp| 65.80 | 67.70 | 69.30 | 70.00 |
> | TPT+CoOp|66.97 |  70.04 |  71.02 |  71.89 |
> | Neural Priming|**71.56**| **71.81** |**72.03** |**72.36** |
>
> **4. In the transductive setting, it is unfair to compare TPT as TPT only relies on a single test sample while the proposed method takes all samples in the test set into account. I think another baseline called TENT should be considered.**
>
> This is a fair point, and we appreciate you pointing us to an additional baseline. We will add this baseline to the paper and have included abbreviated results below.
>
> Comparing with TENT [4] in Transductive Setting
> || ImageNet-V2| ImageNet-Sketch |
> |:-:|:-:|:-:|
> | CLIP| 59.35|57.05 |
> | TENT|59.91|57.40|
> | Transduct. Priming |60.76| 59.97|
>
>
> **5. There are many repeated sentences across the whole paper.**
>
> Thanks for the feedback. We’ve removed redundant sentences throughout the paper, particularly between the introduction and experiments.
>
> **6. Not sure what “3-shot CLIP model” refers to in line 214.**
>
> The 3-shot CLIP model refers to a CLIP model trained with 3 examples per class which are randomly sampled from the training set. In other words, a Neural Primed model with no training examples performs as well as a standard model trained on 3 examples per class. We have revised the paper to make this more clear to readers.
>
> [1] Learning Customized Visual Models with Retrieval-Augmented Knowledge
>
> [2] K-LITE: Learning Transferable Visual Models with External Knowledge
>
> [3] Internet Explorer: Targeted Representation Learning on the Open Web
>
> [4] Tent: Fully Test-time Adaptation by Entropy Minimization

---

> > ### Comment · Reviewer_xxj2 · 2023-08-19
> > **Reply to author rebuttal**
> >
> > Thanks for providing very detailed responses to my concerns. Since most of my concerns have been addressed, I'm happy to raise my rating to weak accept. I hope the authors can incorporate all the changes and new results into the revised version of the paper.

---

> > > ### Author Response · Authors · 2023-08-21
> > > **Thanks for the Update and Feedback**
> > >
> > > Thanks for updating us and raising your rating. We appreciate your constructive comments and believe your recommended experiments and baselines have significantly strengthened the paper.

---

### Author Rebuttal · Authors · 2023-08-09

We thank the reviewers for their thorough, insightful comments and have made revisions based on their feedback. We are glad they found the work well motivated, novel, and the contributions to be of value to the community. We have included a PDF with figures which we reference in our responses. Below are the individual responses to each reviewer which we hope address their concerns. We are happy to engage in further discussion and interested in any additional feedback the reviewers may have.

---

### Decision · Program_Chairs · 2023-09-21

**Decision:**

Accept (poster)

**Comment:**

This paper received diverging reviews. Reviewer 89m2 expressed concerns regarding novelty and experiment, whereas the rest 3 reviewers acknowledge the contribution of this work with unanimous acceptance. Upon reading through the paper and all the discussions, the AC feels that this work does contain enough contributions to be accepted, and the proposed method seems interesting. The authors did a good job addressing the concerns while reviewer 89m2 did not follow up with more questions. In light of this consideration, the AC recommends acceptance.